# Enhancing Spatial Resolution of GNSS-R Soil Moisture Retrieval through XGBoost Algorithm-Based Downscaling Approach: A Case Study in the Southern United States

Qidi Luo ⓘ, Yueji Liang *ⓘ, Yue Guo, Xingyong Liang, Chao Ren ⓘ, Weiting Yue ⓘ, Binglin Zhu and Xueyu Jiang

College of Geomatics and Geoinformation, Guilin University of Technology, Guilin 541004, China; luoqidi@glut.edu.cn (Q.L.); guoyue@nnct.edu.cn (Y.G.); liangxy@glut.edu.cn (X.L.); renchao@glut.edu.cn (C.R.); yueweiting@glut.edu.cn (W.Y.); zhubinglin@glut.edu.cn (B.Z.); xiaojiang@glut.edu.cn (X.J.)
* Correspondence: lyjayq@glut.edu.cn; Tel.: +86-158-783-57721

**Abstract:** The retrieval of soil moisture (SM) using the Global Navigation Satellite System-Reflectometry (GNSS-R) technique has become a prominent topic in recent years. Although prior research has reached a spatial resolution of up to 9 km through the Cyclone Global Navigation Satellite System (CYGNSS), it is insufficient to meet the requirements of higher spatial resolutions for hydrological or agricultural applications. In this paper, we present an SM downscaling method that fuses CYGNSS and SMAP SM. This method aims to construct a dataset of CYGNSS observables, auxiliary variables, and SMAP SM (36 km) products. It then establishes their nonlinear relationship at the same scale and finally builds a downscale retrieval model of SM using the eXtreme Gradient Boosting (XGBoost) algorithm. Focusing on the southern United States, the results indicate that the SM downscaling method exhibits robust performance during both the training and testing processes, enabling the generation of a CYGNSS SM product with a 1 day/3 km resolution. Compared to existing methods, the spatial resolution is increased threefold. Furthermore, in situ sites are utilized to validate the downscaled SM, and spatial correlation analysis is conducted using MODIS EVI and MODIS ET products. The CYGNSS SM obtained by the downscaling model exhibits favorable correlations. The high temporal and spatial resolution characteristics of GNSS-R are fully leveraged through the downscaled method proposed. Furthermore, this work provides a new perspective for enhancing the spatial resolution of SM retrieval using the GNSS-R technique.

**Keywords:** GNSS-R; CYGNSS; SMAP; downscaled; soil moisture



## 1. Introduction

Soil moisture (SM) plays a pivotal role in many natural phenomena and processes. For instance, it directly affects crop growth and can be a significant factor in natural disasters such as land degradation, floods, and landslides [1]. These issues have profound impacts, including on food security and the stability of ecological environments, making accurate and real-time monitoring of SM particularly important. However, traditional SM detection methods have notable limitations. These methods primarily rely on direct measurements from ground detectors or meteorological stations, which means they require substantial human and material resources and are time-consuming [2]. Moreover, due to the limitations of these methods, they cannot achieve large-scale, efficient, and low-cost SM retrieval. For vast areas and complex terrains, their detection performance is severely limited. Fortunately, the advent of remote sensing technology provides a new avenue to address this issue. Remote sensing technology can use satellites or drones to monitor the ground from the air, avoiding the difficulties of ground detection and thus achieving large-scale SM retrieval [3,4]. In fact, the European Space Agency (ESA) and the National Aeronautics and Space Administration (NASA) have launched the Soil Moisture and Ocean Salinity (SMOS) satellite [5] and the Soil Moisture Active Passive (SMAP) mission [6] for

SM retrieval. Both missions can achieve global SM retrieval with a spatial resolution of about 40 km, and they can revisit the globe every 2–3 days. However, while remote sensing technology and related satellite missions such as SMOS and SMAP provide global SM retrieval capabilities, the resolution of these products is relatively low, making them more suitable for large-scale applications. For medium- and small-scale applications that require more detailed observations, such as irrigation management in farmland or flood warnings in specific areas, these methods may not meet the needs.

The technique of Global Navigation Satellite System-Reflectometry (GNSS-R) represents a novel type in the field of remote sensing. Its internal L-band signal source is adequate and exhibits high penetration capabilities for vegetation, soil, snow, etc. It is capable of all-weather, all-day observation and has excellent potential for SM retrieval [7–9]. GNSS-R receivers are generally installed on the ground or on aircraft. Although they have an excellent detection accuracy, the monitoring range limits its ability to achieve a wide range of SM retrieval [10]. CYGNSS was successfully launched in 2016, with a revisit cycle of 2.8 (median) and 7.2 (average) hours [11], providing ample data for SM retrieval by GNSS-R technique. Thus, using the GNSS-R technique to retrieve SM has become a hot research topic in recent years. Chew et al. [12] showed that there is a strong linear relationship between the surface reflectance of CYGNSS and SMAP SM, and a global SM product with a resolution of 36 km was produced through linear method. Ruf [13] proposed that SMAP SM can be supplemented by using the relative signal-to-noise ratio (rSNR) of CYGNSS to SM retrieval. Al-Khaldi et al. [14] considered that vegetation and surface roughness would affect SM. They proposed a method for CYGNSS SM retrieval through time series. A global SM product of $0.2° \times 0.2°$ was finally generated. Considering that the terrain, vegetation, and surface roughness have an impact on the GNSS signal, the relationship between the signal and SM is relatively complex and nonlinear. Machine learning has been frequently used in the study of CYGNSS SM retrieval because of its great advantages in handling nonlinear situations. Eroglu et al. [15] combined CYGNSS observables with in situ sites observations, Vegetation Water Content (VWC), Normalized Vegetation Index (NDVI), and topography features. Finally, a daily SM product with a resolution of 9 km was generated using the Artificial Neural Network (ANN). Senyurek et al. [16] obtained the daily SM of the United States with a resolution of 36 km using CYGNSS and in situ site observations based on machine learning algorithms. The results showed that the prediction effect of Random Forest (RF) was the best, with an RMSE of 0.052. Jia et al. [17] pre-classified land cover types and used the eXtreme Gradient Boosting (XGBoost) method for SM retrieval. Compared with the accuracy of SM retrieval without pre-classification, there was an improvement, with an RMSE of 0.052.

However, the SM products obtained from the aforementioned microwave remote sensing data have a coarse resolution, which limits their utility in medium- and small-scale hydrological and agricultural applications. Zhan et al. [18] first introduced an empirical polynomial for downscaling, marking an initial exploration of effective strategies to address this issue. Subsequently, Chauhan et al. [19] improved upon Zhan's method, enhancing its performance. In this empirical polynomial downscaling method, high-resolution SM is expressed as a polynomial function of surface temperature, plant index, and surface reflectance derived from brightness temperature data. This innovative method provides a fresh perspective for tackling the downscaling of SM. Piles et al. [20] further optimized this downscaling polynomial fitting method. Their improvement replaced surface reflectance in the polynomial equation with coarse-resolution brightness temperature data, making the method more flexible and efficient in handling practical problems. Moreover, this polynomial fitting downscaling method has been widely applied in the downscaling of various SM products, such as SMOS and AMSR-E, and also in various high-resolution remote sensing image products, such as MODIS and MSG-SEVIRI. This has been confirmed by many scholars [21–26]. Their research further validates the practicality and broad application value of this method. In order to retrieve daily SM at a 9 km resolution, Das et al. [27] downscaled the coarse-resolution (approximately 40 km) SMAP L-band

brightness temperature data using the high-resolution (1–3 km) L-band Synthetic Aperture Radar (SAR) backscatter observations. Based on artificial intelligence techniques including Support Vector Machines, Artificial Neural Networks, and Associated Vector Machines, Srivastava et al. [28] fused MODIS surface temperature with SMOS SM and enhanced the spatial resolution of SMOS SM by using downscaling methods. The factor used to represent the high-resolution state of SM plays a crucial role in determining the accuracy of the downscaled SM. The downscaled SM has higher accuracy compared to the original coarse-resolution SMOS and AMSR-E SM, with the *R* rising from 0.27 to 0.96 [29]. Compared to the observed data, the accuracy of the downscaled SM has improved relative to the products of SMOS and AMSR-E [30]. This means that downscaling methods could be attempted to provide high-resolution SM for products such as SMAP, SMOS, AMSR-E, and NASA-USDA.

The aforementioned research demonstrates both the significant advantages of using GNSS-R technique for SM retrieval and the notable effects of using downscaling methods to enhance the spatial resolution of SM products. However, no studies have yet used the downscaling method to improve the spatial resolution of GNSS-R technique. At present, the spatial resolution achieved by SM retrieval based on spaceborne GNSS-R is limited (up to 9 km). Spatial downscaling of microwave SM is a crucial strategy. It addresses the pressing need for higher spatial resolution SM data, which is essential for local hydrological or agricultural applications. Therefore, this paper proposes a method for constructing a SM downscaling model. This method aims to fuse the CYGNSS observables and auxiliary variables with SMAP SM (36 km) products, forming a nonlinear relationship at the same scale. Finally, a downscaling model will be built based on the XGBoost algorithm to retrieve SM with a spatial resolution of 3 km. In the end, the SM retrieval using GNSS-R technique is successfully spatially downscaled, improving the spatial resolution of SM retrieval.

## 2. Materials and Methods

### 2.1. Study Area

The study area is located in the southern United States, characterized by diverse terrains and rich ecosystems. The region experiences a subtropical humid climate, with an average annual precipitation of approximately 834.45 mm, contributing to the area's rich biodiversity and thriving ecosystems. Geographically, the study area exhibits significant variations in altitude, ranging from −88 m to 4277 m, with an average altitude of 1778 m. The terrain generally features higher elevations in the west and lower in the east. This variation in terrain provides excellent conditions for studying the relationship between SM and environmental factors such as terrain and climate. Ecologically, the primary land cover types in the study area are grasslands and tropical savannas, collectively accounting for 55% of the total area. Additionally, a considerable portion of the western region is covered by open shrublands, making up 13% of the total area. The elevation and land cover types of the study area are shown in Figure 1.

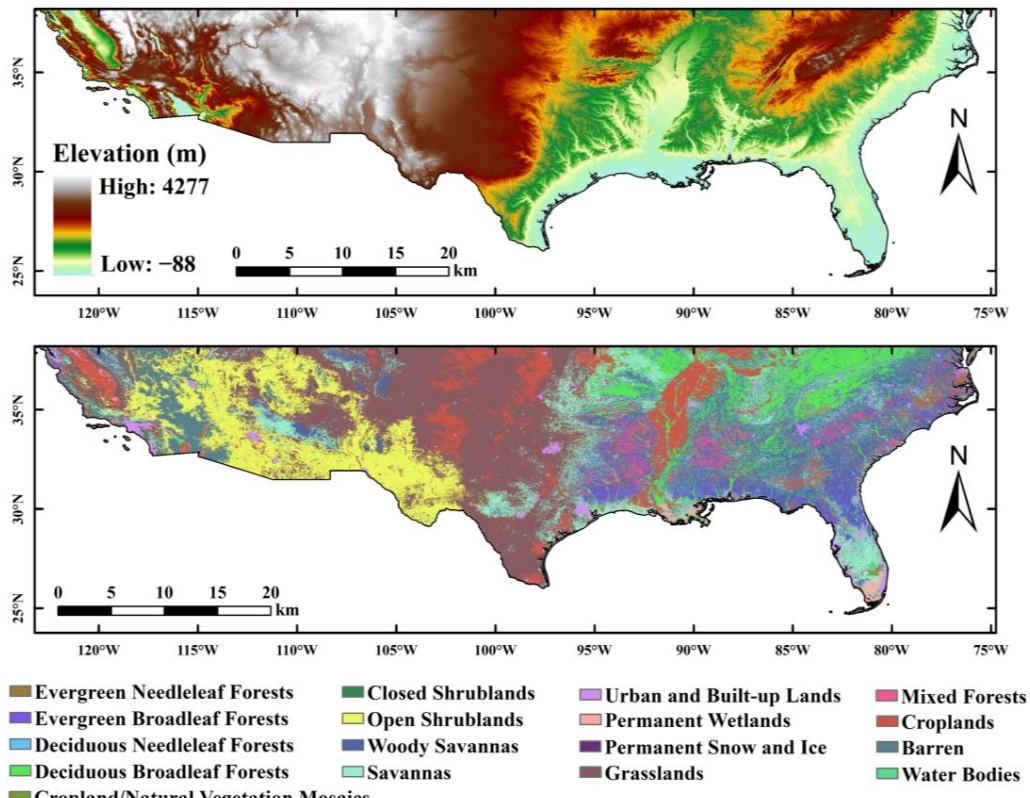

**Figure 1.** DEM and land cover type map of the study area.

### 2.2. Cyclone Global Navigation Satellite System

As a component of NASA's Earth System Science Pathfinder project, the Cyclone Global Navigation Satellite System (CYGNSS) was launched on 15 December 2016. The observatories are composed of eight microsatellites. They offer almost uninterrupted coverage of the Earth due to their orbit inclination of approximately 35° to the equator. This positioning results in an average revisit time of 7 h and a median revisit time of 3 h. This inclination allows CYGNSS to cover an observational range from 38°N to 38°S. Therefore, we selected the southern part of the United States as the study area (CYGNSS observables cannot cover the entire US).

The objective of this study is to retrieve SM within a specific region. To achieve this, we utilized the CYGNSS Level-1 (L1) version 2.1 product, with data sourced from the Physical Oceanography Distributed Active Archive Center (PO.DAAC, https://podaac.jpl.nasa.gov/, accessed on 1 April 2023). The primary goal of CYGNSS is to enhance understanding and prediction of tropical cyclone intensity by leveraging signals from the Global Navigation Satellite System (GNSS). The core component of this system is the Delay Doppler Mapping Instrument (DDMI), whose main task is to generate Delay Doppler Maps (DDMs) [31]. DDMs represent the received surface power of each observed specular reflection point through a series of time delays and Doppler frequencies, measured on a bin-by-bin basis. In other words, they provide a two-dimensional representation of the reflection characteristics of GNSS signals. These characteristics are influenced by factors such as SM and vegetation cover, and can therefore be used to infer SM. It is important to note that the DDMI initially measures in uncalibrated "counts", which have a linear relationship with the total signal power it processes. The total signal power includes thermal radiation from the Earth and the DDMI itself, as well as GPS signals scattered from the land surface. However, during the Level-1A calibration process, each bin in the DDM converts these raw counts into watts, allowing for a more intuitive understanding and analysis of the data. The CYGNSS observables used in this paper cover the period from 1 January to 31 December 2019.

The surface reflectivity can be estimated through a variety of methods with various coherence and incoherence assumptions using the observables in the L1 data [15,32,33]. In water accumulation areas such as lakes, rivers, and wetlands, low surface roughness leads to dominant coherent scattering in forward scattering. Even with higher SM, coherent forward scattering remains strong due to water. However, GPS signals interacting with vegetation introduce some incoherent components. Higher SM regions show a stronger signal intensity due to a relatively higher SNR compared to lower SM areas. Thus, in this paper, we adopted the approach proposed by Rodriguez-Alvarez et al. [32] to calculate reflectivity, under the assumption that the observed GNSS-R signal is predominantly made up of coherent reflections. This involves using the BRCS (denoted as '*brcs*' in CYGNSS L1) and the range terms to calculate the reflectivity ($\Gamma_{RL}(\theta_i)$) as:

$$\Gamma_{RL}(\theta_i) = (\frac{4\pi}{\lambda})^2 \frac{P_{RL}^{coh}(rst + rsr)^2}{P_t G_t G_r} \tag{1}$$

where $P_{RL}^{coh}$ represents the dual base radar coherent receive power. The subscripts $R$ and $L$ stand for the right circularly polarized GNSS transmit antenna and the left circularly polarized GNSS-R antenna, respectively. The GNSS signal wavelength is denoted by $\lambda$. *rst* and *rsr* refer to the distances from the specular reflection point to the GNSS transmitter and the GNSS-R receiver, respectively. $P_t$ signifies the peak power of the transmitting GNSS signal. $G_t$ and $G_r$ are the gains of the transmitting and receiving antennas, respectively. Lastly, $\Gamma_{RL}(\theta_i)$ is the surface reflectance at an incidence angle of $\theta_i$.

Leading Edge Slope (LES) and Trailing Edge Slope (TES) are indicators associated with coherent or incoherent scattering conditions. An increase in the incoherent reflection component within the reflected signal typically results in a corresponding increase in the absolute values of both LES and TES. Following the methodologies presented by Carreno-Luengo et al. [34] and Rodriguez-Alvarez et al. [32], LES and TES can be calculated as follows:

$$\text{LES} = \frac{\Gamma_m - \Gamma_{m-3}}{3\Delta} \tag{2}$$

$$\text{TES} = \frac{\Gamma_{m+3} - \Gamma_m}{3\Delta} \tag{3}$$

where $\Gamma_m$ represents the peak reflectivity at the reflection point, $\Gamma_{m-3}$ is the reflectivity at the third point before the reflection point, $\Gamma_{m+3}$ is the reflectivity at the third point after the reflection point, and $\Delta$ stands for the delay resolution of the Doppler delay map, which is 0.2552 chips.

DDM_SNR is one of the most basic variables in CYGNSS observables. When the value of SM increases in the same area, the difference between the corresponding values of DDM_SNR also increases. Therefore, DDM_SNR is added to the model as a factor affecting the SM retrieval. For SM retrieval in the machine learning framework, the derived reflectivity, together with LES, TES, and DDM_SNR, are used as the input layer characteristics of CYGNSS observables.

### 2.3. Soil Moisture Active Passive Data

The reference products utilized in this study primarily originate from the SMAP satellite, launched by NASA in 2015. The primary mission of this satellite is to monitor global surface SM and freeze-thaw states, aiming to gain a deeper understanding and knowledge of the Earth's surface water cycle, climate change, ecosystem dynamics, and the impact of human activities. The SMAP satellite employs an L-band radiometer for its observations, a device capable of penetrating clouds and most vegetation to directly measure microwave radiation from the ground, thereby inferring SM and freeze-thaw states. The SMAP satellite revisits each location every 2–3 days, offering a very short global coverage cycle. Notably, the SMAP satellite carries out two types of observations: ascending

(6:00 a.m.) and descending (6:00 p.m.) [35]. This design allows for comparisons and analyses at different times for the same location, providing more comprehensive information.

The data used in this study is obtained from the National Snow and Ice Data Center (NSIDC, https://nsidc.org/, accessed on 15 April 2023). We selected the SMAP Level-3 (L3) Radiometer Global Daily 36 km EASE-Grid Soil Moisture (Version 8, SPL3SMP) as the reference product. This product offers daily estimates of global land surface conditions. The data derived from SMAP's L-band are resampled to a global, cylindrical, 36-km Equal-Area Scalable Earth Grid. The data period is from 1 January to 31 August 2019, providing ample samples for our study.

The original format of the SMAP product is HDF5. In this study, we use the HEG tool (HDFEOS To GeoTIFF Conversion Tool) to convert it into an easily processed Geotiff data format.

### 2.4. International Soil Moisture Network

In this paper, the in situ SM observations from the ISMN sites [36] are used to validate the CYGNSS SM data predicted by the downscaling model. Globally, the ISMN has set up more than 50 SM monitoring networks that are either operational or experimental. These networks provide a unified in situ SM database on a global scale, with a standardized data format and pre-processing quality flags [37]. The majority of sites that offer time and space co-located with CYGNSS observables are located in North America. Consequently, we selected 78 available sites within the spatial coverage of CYGNSS for our study (Figure 2). These sites primarily belong to the Soil Climate Analysis Network (SCAN), the U.S. Climate Reference Network (USCRN), and the Snow Telemetry Network (SNOTEL). The hourly SM data from the ISMN was processed by filtering it with the provided quality mark (marked with a "G" for "good") and subsequently converting it into daily averages. The surface SM data utilized was at a depth of 5 cm, aligning with the penetration depth of L-band microwave signals. For a comprehensive overview of ISMN, readers can refer to [36,38]. The ISMN dataset can be accessed publicly (http://ismn.geo.tuwien.ac.at, accessed on 20 April 2023).

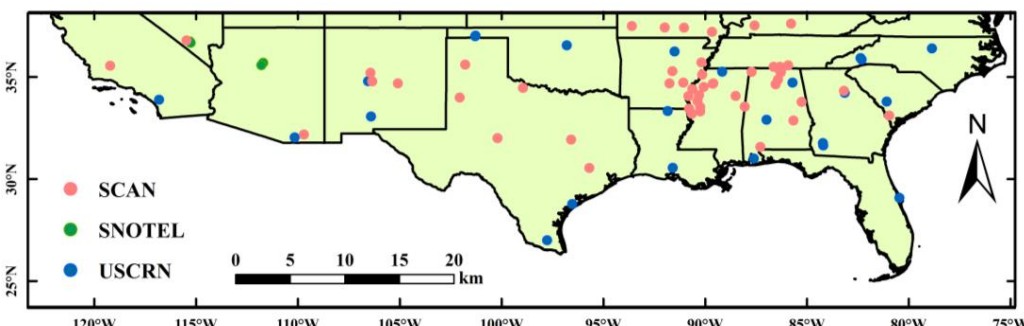

**Figure 2.** Geographic distribution of International Soil Moisture Network (ISMN) sites in the study area.

### 2.5. Auxiliary Data

According to existing research, SM is influenced by additional variables in addition to rainfall, including elevation, land cover type, annual accumulated days, Normalized Difference Vegetation Index (NDVI), and latitude and longitude information of satellite sampling points [31,39]. These factors are typically used as auxiliary variables in downscaling methods [40–43].

Topography, as a significant non-living factor, greatly influences the variability of soil hydrothermal resources. The differences in elevation directly impact the spatial redistribution of solar radiation and rainfall. Therefore, in our downscaling model, we incorporated altitude as the topographic variable. The source of altitude data is the Shuttle Radar Topography Mission (SRTM) [44]. The influence on SM varies with different types

of land cover, as they have different effects on the storage and release of moisture. The land cover type data used in this paper is based on the International Geosphere Biosphere Programme (IGBP) [45] land cover map derived from the MODIS. NDVI is widely used to assess vegetation growth, drought conditions, and ecological environments. Since NDVI exhibits a high sensitivity to factors such as vegetative cover and SM content, it is also used for retrieving SM and vegetation covering [46]. The NDVI product is calculated from the daily 250 m product provided by MODIS (MOD09GQ). The precipitation plays a significant role in vegetation growth and has a strong impact on SM. Precipitation affects SM as it comes into contact with the soil, and there is a positive correlation between SM and precipitation. Therefore, we include precipitation as an input variable in the model. The daily average precipitation in the study area is obtained through the Climate Hazards Group InfraRed Precipitation with Station data (CHIRPS) project. Table 1 summarizes the fundamental characteristics of the auxiliary variables used in this paper.

**Table 1.** Overview of data utilized for the downscaling process in this paper.

| Datasets | Variables | Temporal Resolution (Day) | Spatial Resolution | Time |
|---|---|---|---|---|
| MOD09GQ | NDVI | 1 | 250 m | 1 January–31 December 2019 |
| MCD12Q1 | Land cover | - | 500 m | 2019 |
| SRTM | DEM | - | 30 m | 2019 |
| CHIRPS | Precipitation | 1 | 0.5° | 1 January–31 December 2019 |
| - | Lon, Lat | 1 | - | 1 January–31 December 2019 |
| - | Doy | 1 | - | 1 January–31 December 2019 |

## 3. Soil Moisture Downscaling Framework

### 3.1. Random Forest (RF)

Ho et al. [47] first proposed the concept of Random Forest (RF) in 1998; then, Breiman et al. [48] systematically developed it in 2001. RF is a collective model constructed on the foundation of decision trees. It is implemented through the Bagging concept of ensemble learning, aiming to solve the problem of overfitting that is common in single decision tree algorithms. The decision tree is a fundamental component. Due to its significant advantages in handling high-dimensional feature data and large datasets, RF is widely used in multivariate regression problems. Compared with ordinary decision trees, RF makes some improvements in the process of building decision trees. During the generation of a regular decision tree, the optimal feature among all sample features on the node is used. However, the RF algorithm randomly selects a certain number of attribute features when generating a decision tree, and then picks the optimal feature from these randomly selected features to construct the decision tree. The decision trees built using RF have different structures. They will not lead to overfitting due to the addition of more trees, but instead produce a limited value of generalization error. This approach not only reduces fitting errors but also avoids repetitive learning, which helps to enhance the predictive performance of the final model and improve its generalization ability.

The process of the random forest algorithm is: (1) Performing $n$ random samplings on the training dataset, each time taking $m$ samples, to obtain a subset of data $S_n = \{(x_1, y_1), (x_2, y_2), (x_3, y_3), \cdots, (x_m, y_m)\}$ containing $m$ samples. (2) Using these sub-datasets to train $n$ weak prediction models $f_n(x)$ separately. (3) When training decision tree model nodes, a subset of feature samples is selected from all samples. Then, the optimal feature for splitting the decision tree is chosen from this randomly selected subset of feature samples. (4) The results of the various weak prediction models are consolidated according to the specific problem at hand. For regression functions, the final output is the arithmetic average of all the weak prediction models.

### 3.2. eXtreme Gradient Boosting (XGBoost)

The eXtreme Gradient Boosting (XGBoost) method, proposed by Chen et al. [49], is also an ensemble learning method based on gradient boosting machines. Similar to RF, XGBoost is a learner based on Classification and Regression Trees (CART). It implements ensemble learning of multiple CART trees by optimizing the traditional Gradient Boosting Decision Tree (GBDT). It can be used to solve various machine learning problems, including classification and regression. While each tree in the RF algorithm is trained in parallel, the decision trees in XGBoost are not mutually independent. The construction process of the XGBoost model is as follows: First, an initial tree is built using the training set for model training, which results in residuals between the model's predicted and actual values. Then, during each iteration, a tree is added to fit the residuals from the model's previous prediction until the model's learning process is terminated. Ultimately, this forms an iterative residual tree collection, an ensemble of numerous tree models. The predicted value can be calculated as follows:

$$\hat{y}_i = \sum_{k=1}^{K} f_k(x_i) \tag{4}$$

where $\hat{y}_i$ represents the final model prediction value, $K$ represents all the built CART trees, $x_i$ represents the features of the $i$ sample, and $f_k(x_i)$ represents the prediction value of the $k$ tree. The objective function calculation formula for XGBoost is shown in Equation (5):

$$O_{bj} = \sum_{i=1}^{m} l(\hat{y}_i, y_i) + \sum_{k=1}^{K} \Omega(f_k) \tag{5}$$

where $m$ represents the total amount of sample data imported into the $k$ tree. The first term is the loss function, which measures the error between the true value $y_i$ and the predicted value $\hat{y}_i$. The second term is the regularization term, used to control the model's complexity and prevent overfitting. The complexity of each tree is defined as:

$$\Omega(f) = \gamma T + \frac{1}{2}\lambda ||w||^2 \tag{6}$$

where $\gamma$ represents the difficulty of node splitting, $T$ represents the number of leaf nodes, $\lambda$ is the L2 regularization coefficient to prevent overfitting, and $w$ is the modulus of the leaf node vector.

### 3.3. Light Gradient Boosting Machine (LGBM)

Developed by Microsoft Research in 2017, Light Gradient Boosting Machine (LGBM) stands as one of the most effective and advanced machine learning algorithms [50]. LGBM has evolved from the boosting regression algorithms. It employs a histogram-based algorithm, storing continuous features into discrete bins. The use of a histogram-based method accelerates the training speed and reduces memory usage. Additionally, LGBM utilizes the leaf-wise tree growth algorithm. The growth process involves choosing the leaf with the highest delta loss. This contrasts with many boosting algorithms (such as XGBoost) that use a level-wise approach. Although a level-based approach ensures a consistent number of leaves at each level, the leaf-wise strategy leads to a different number of leaves at each respective level. This approach helps LGBM achieve lower loss. The main process of the LGBM algorithm is shown in Equation (7):

$$F_n(x) = \alpha_0 f_0(x) + \alpha_1 f_1(x) + \cdots + \alpha_n f_n(x) \tag{7}$$

where the classifier begins with $n$ decision trees, and the weight assigned to the training samples is $\frac{1}{n}$. The weak classifier $f(x)$ and its weight $\alpha$ are determined. The process continues, with the classifier adjusting the weights until it arrives at the final classifier, denoted as $F_n(x)$. In summary, the main goal of the LGBM algorithm is to improve training

efficiency and accuracy through feature parallelization and a histogram-based decision tree algorithm. It also uses gradient boosting methods to continuously optimize the model, thereby achieving better classification and regression results.

### 3.4. Genetic Algorithm, Back Propagation (GA-BP)

The Back Propagation (BP) neural network, a classic model in ANN (Artificial Neural Networks), was first proposed by Hecht-Nielsen et al. [51]. This network comprises an input layer, hidden layers, and an output layer, with neurons connecting each layer. The output of a neuron depends on its input values, activation function, and threshold. The BP neural network consists of two steps: forward propagation of information and backward propagation of errors. Although the BP neural network has excellent self-learning, adaptability, and self-organization capabilities and can effectively handle non-linear problems, it has some limitations: Firstly, in order to reduce error and improve accuracy, an appropriate number of neurons in the hidden layer need to be selected. However, there is a lack of a clear method for this selection. Secondly, the BP neural network randomly generates initial weights and thresholds. This results in adaptive and global approximation processes that are time-consuming, thereby slowing the network's convergence rate. Lastly, the use of gradient descent by the BP neural network can often lead to it becoming trapped in local minima.

The Genetic Algorithm (GA) is a global optimization probabilistic search method based on the principles of biological inheritance and evolution [52]. The GA mainly includes three operations: (1) Selection operation: The probability of an individual entering the next generation population is determined based on the fitness value. The higher the fitness, the greater the chance of inheritance. (2) Crossover operation: This is a key part of the algorithm. Two individuals are selected from the population, and a portion of their genes are exchanged to produce more optimal individuals in the new generation. (3) Mutation operation: An individual is randomly selected from the population, and a mutation is performed at a certain locus of its chromosome to produce a more optimal individual. Combining crossover and mutation operations can achieve optimal search performance. The GA has the characteristics of global search and parallel computation, but it lacks learning ability. The application of GA can optimize the BP neural network. This combines the GA's global search traits with the BP's learning and non-linear mapping abilities. As a result, the network's output accuracy improves.

### 3.5. Performance Metrics and Evaluation

The performance of models and the retrieval accuracy of downscaled SM are evaluated in this paper using three indicators: Mean Absolute Error (MAE), Root Mean Square Error (RMSE), and correlation coefficient (*R*). MAE is calculated as the average of the absolute differences between each observation and the mean. This method circumvents the issue of error cancellation, thereby providing a more accurate representation of the actual prediction error magnitude. RMSE is commonly used as a standard to measure the prediction results of machine learning models. The *R* can be used to measure the degree of correlation between two variables. The calculation formulas for the three indicators are as follows:

$$\text{MAE} = \frac{1}{n}\sum_{i=1}^{n}|X_i - Y_i| \tag{8}$$

$$\text{RMSE} = \in \sqrt{\frac{1}{n}\sum_{i=1}^{n}(X_i - Y_i)^2} \tag{9}$$

$$R(X,Y) = \frac{Cov(X,Y)}{\sqrt{Var[X]Var[Y]}} \tag{10}$$

where $n$ represents the amount of data used for modeling, $X$ is the reference value of SM, and $Y$ is the retrieved value of SM. These three values are crucial for us to evaluate the

prediction accuracy of the model. Among them, $X$ is the known true value, and $Y$ is the value predicted by our models. To more accurately evaluate the performance of the model, we introduce several key statistical indicators. Among them, $Cov(X, Y)$ represents the covariance of $X$ and $Y$, which describes the degree of joint variation of $X$ and $Y$. At the same time, $Var[X]$ is the variance of $X$, and $Var[Y]$ is the variance of $Y$. These two indicators describe the range of variation of $X$ and $Y$, respectively. These three indicators jointly evaluate the performance and prediction accuracy of the model. Covariance describes the correlation between the model's predicted values and the true values, while variance shows the dispersion of the data. The changes in these data directly affect the predictive ability of the model.

*3.6. Downscaling Process*

This study employed four ML techniques, including RF, XGBoost, LGBM, and GA-BP, to downscale CYNGSS SM retrieval to 3 km, respectively. Each of the proposed downscaling methods operates on a common principle: they create a statistical link between CYGNSS, geospatial variables (such as elevation and land cover type), land-surface variables (such as NDVI and precipitation), and SMAP SM at a coarse resolution of 36 km. In addition, SMAP SM was used as the reference value of SM, predicted by the downscaling model. Finally, the output covariates of the input variables were linked by using the following equation:

$$SM = f(\rho_1, \rho_2, \rho_3, \ldots, \rho_n) + \varepsilon \tag{11}$$

where SM represents the downscaled SM data, which is determined by the regression function of the machine learning models (RF, XGBoost, LGBM, and GA-BP), $\varepsilon$ is the model retrieval error, $\rho_1, \rho_2, \rho_3, \ldots, \rho_n$ represent the input covariates (i.e., SNR, SR, LES, TES, NDVI, DEM, land cover type, and precipitation). The total number of predictors is represented by $n$. The steps of the downscaling method mentioned above can be briefly summarized as follows:

1.  Aggregation: The training procedure is carried out on the 36-km grid of SMAP SM. To maintain consistency with the spatial resolution of SMAP SM, the high-resolution CYGNSS observables and auxiliary variables (i.e., predictive factors) are aggregated to a 36 km scale using a simple arithmetic averaging method. It is worth noting that the theoretical resolution of the CYGNSS dataset used in this study is $7 \times 0.5$ km, while the SMAP product resolution is 36 km. This means that multiple CYGNSS observation points inevitably exist within the same SMAP grid. To address this, we average the CYGNSS sample points within the SMAP grid during the spatial matching process. Figure 3 presents the daily count of CYGNSS sampling points within the study area, as well as the counts after matching with SMAP. Specifically, from January to August, the counts correspond to the match with SMAP's 36-km grid; from September to December, the counts result from matching with the resampled 3-km grid of SMAP.
2.  Model building: The aggregated data is divided into 70% for training and 30% for testing. SMAP SM is used as the response variable, and CYGNSS observables and auxiliary variables are used as input variables to train the four models: RF, XGBoost, LGBM, and GA-BP.
3.  Resampling: The CYGNSS observables and auxiliary variables are resampled to a high resolution of 3 km using the nearest neighbor method. Subsequently, a spatial connection is established between them.
4.  Model application: The resampled 3-km high-resolution CYGNSS observables and auxiliary variables were input to the downscaling models to obtain the downscaled CYNGSS SM with a spatial resolution of 3 km.
5.  Model evaluation: Once the most optimal downscaling models were determined based on the lowest RMSE as a benchmark, these models were then evaluated for accuracy using the testing set. Then, the 3-km downscaled SM obtained was validated

using in situ data. Spatial analysis of the downscaled CYGNSS SM is conducted using MODIS EVI and MODIS EV products.

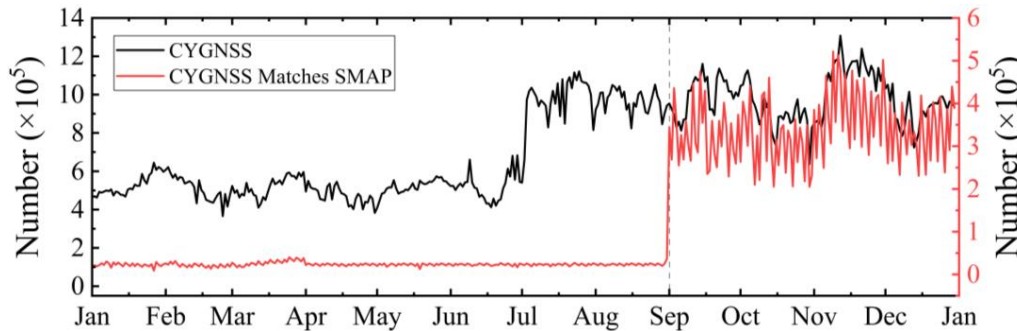

**Figure 3.** Daily sampling counts of CYGNSS and their corresponding matched counts with SMAP.

The experimental process is based on the assumption that the spatial scale relationship among SMAP SM, CYGNSS observables, and auxiliary variables maintains consistency. In other words, the relationship models established at a coarse resolution are still applicable at a high resolution [39,53,54]. The above experimental process is shown in Figure 4.

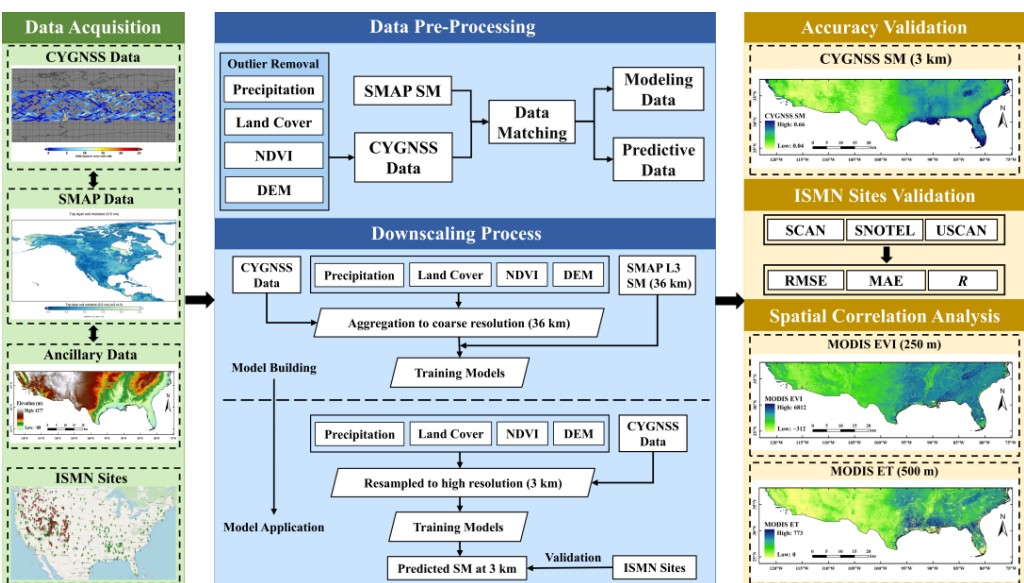

**Figure 4.** Flow chart of downscaling procedure.

## 4. Results

### 4.1. Models Evaluation

Following the methodologies outlined in Section 3, this paper constructs four CYGNSS SM downscaling models and adjusts the hyperparameters for the RF, XGBoost, LGBM, and GA-BP models. Hyperparameters are parameters given in advance in neural networks or machine learning to control the learning process of the model. The appropriate selection of hyperparameters is crucial for the predictive performance of the model and can also prevent the occurrence of overfitting or underfitting. The common hyperparametric methods are Grid search, Random search, and Bayesian optimization. This paper uses the Grid Search method for hyperparameter adjustment. Although Grid Search requires a longer runtime compared to the other two hyperparameter selection methods, it is a more exhaustive search method that ensures the best hyperparameter combination is found within the given parameter range. The final hyperparameter adjustment results are shown in Table 2.

**Table 2.** Results of hyperparameter adjustment for the four models.

| Model | Hyperparameters |
|-------|-----------------|
| GA-BP | popu = 50, learning_rate = 0.001, epochs = 100, n_hidden layer = 10 |
| RF | n_estimators = 100, max_depth = 6, max_leaf_nodes = None, min_samples_leaf = 1, min_samples_split = 2 |
| XGBoost | booster = tree, max_depth = 8, min_child_weight = 1, leaing_rate = 0.25, n_estimators = 100, subsample = 0.9, colsaple_bytree = 0.6, gamma = 0 |
| LGBM | learning_rate = 0.09, n_estimators = 100, min_samples_gain = 0.1, max_depth = 6, num_leaves = 50, subsample = 0.8, colsample bytree = 0.8 |

Through this process, we found the best combination of hyperparameters for each model to ensure that the CYGNSS SM downscaling models have high predictive performance. In the subsequent analysis, we will use these optimal hyperparameter combinations to train the model and evaluate its performance. To preliminarily assess the performance of the four models, this paper uses the method of ten-fold cross-validation for comparative analysis. The dataset uses SMAP SM (36 km) as the reference value, and four CYGNSS parameters, including SR, SNR, LES, and TES, as well as the auxiliary variables described in Section 2. This paper selects the coarse resolution data (36 km) from January to August 2019 to construct the downscaling model, yielding a total of 303,354 samples. For the prediction dataset, we use high-resolution data (3 km) from September to December, which provides a total of 4,123,129 samples. The ten-fold cross-validation accuracy of the four models and the running time of the models are shown in Table 3 and Figure 5.

**Table 3.** Summary of the overall accuracy of the ten-fold cross-validation of the four models.

| Name | Time t/s | Model Training RMSE cm³/cm³ | MAE cm³/cm³ | R - | Model Testing RMSE cm³/cm³ | MAE cm³/cm³ | R - |
|------|----------|------|------|------|------|------|------|
| GA-BP | 46.16 | 0.071 | 0.055 | 0.834 | 0.072 | 0.055 | 0.831 |
| RF | 5712.91 | 0.011 | 0.007 | 0.996 | 0.031 | 0.021 | 0.969 |
| XGBoost | 25.90 | 0.037 | 0.027 | 0.958 | 0.038 | 0.028 | 0.955 |
| LGBM | 6.85 | 0.045 | 0.033 | 0.937 | 0.045 | 0.034 | 0.935 |

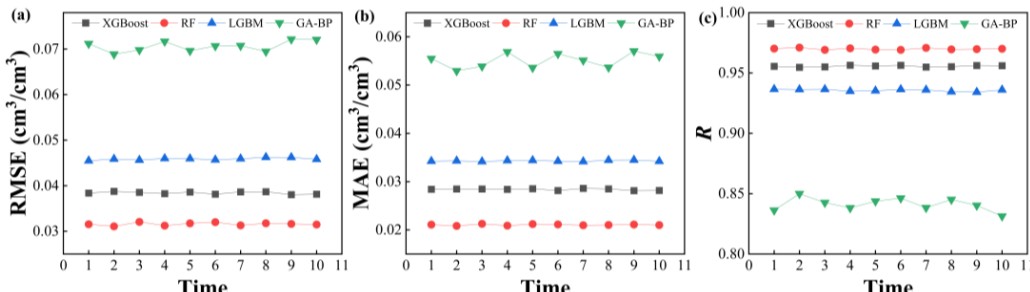

**Figure 5.** Summary of the accuracy of the ten-fold cross-validation of the four models. (**a**) RMSE; (**b**) MAE; (**c**) *R*.

Table 3 and Figure 5 present the accuracy of the ten-fold cross-validation of the four models and their execution times. In terms of execution time, the RF model took the longest, reaching 5712.91 s. This could be due to the fact that the RF model needs to generate a large number of decision trees during the training process and carry out complex voting and averaging operations, thus taking a longer time. The execution time of the GA-BP model was 46.16 s, significantly shorter than the RF model. However, the predictive performance

of the GA-BP model was not satisfactory, with an RMSE of 0.072, an MAE of 0.055, and an *R* of 0.831. These metrics indicate that the GA-BP model has relatively low accuracy and stability in predicting SM. The XGBoost model had a shorter execution time of 25.9 s. Its predictive performance was relatively good, with an RMSE of 0.038, an MAE of 0.028, and an *R* of 0.955. These metrics indicate that the XGBoost model has high accuracy and stability in predicting SM. The LGBM model had the shortest execution time, only 6.85 s, although its predictive performance was not as good as the RF and XGBoost models, with an RMSE of 0.045, an MAE of 0.034, and an *R* of 0.935. Nevertheless, these metrics still indicate that the LGBM model has acceptable predictive performance. Therefore, considering both execution time and predictive performance, the LGBM model performs best in terms of time efficiency, but its predictive performance is slightly worse than the RF and XGBoost models. Although the RF model takes the longest time, it has the best predictive performance. The XGBoost model performs well in both execution time and predictive performance. Although the GA-BP model has a shorter execution time, it has the worst predictive performance.

Figure 6 presents the performance of the downscaling models XGBoost, RF, LGBM, and GA-BP, which were constructed based on CYGNSS and auxiliary variables. The CYGNSS SM predictions at a coarser resolution (36 km) were compared with the SMAP SM predictions at the same resolution using the scatter plots for each model. It can be seen that the XGBoost and RF models perform well, exhibiting strong consistency between the CYGNSS SM and SMAP SM in both the training and testing set. The *R* for the training set is 0.95 and 0.99, respectively, while, for the testing set, they are both 0.95. However, the GA-BP model shows less satisfactory retrieval results, with an *R* of 0.84 for both the training and testing set. When comparing the RMSE of the models mentioned, XGBoost and RF models clearly outperform, with an RMSE of 0.038 and 0.012 for the training set, and 0.039 and 0.033 for the testing set. In contrast, the GA-BP and LGBM models show a higher RMSE, with 0.069 and 0.045 for the training set, and 0.070 for both in the testing set. The RF model exhibits the lowest MAE, with 0.008 for the training set and 0.022 for the testing set. However, the GA-BP and LGBM models show a higher MAE, with 0.054 and 0.033 for the training set, and 0.054 and 0.055 for the testing set, respectively.

The results indicate that the downscaling models built on RF and XGBoost outperform the models constructed using LGBM and GA-BP. Overall, the RF and XGBoost downscaling models demonstrate superior correlation and less error compared to the other models. This may be due to the robustness and unpredictable nature of the RF and XGBoost algorithms. When dealing with numerous variables at once, these techniques are intended to avoid overfitting. However, compared to the RF model, the XGBoost model achieves high accuracy with less time. Therefore, we mainly focus on downscaled SM from XGBoost model in the following sections.

Figure 7 presents the importance scores for various variables in retrieving outcomes with the XGBoost model. Among all input variables, the greatest impact on both the 36 km and 3 km resolution is seen with land cover and DEM. In particular, NDVI has a more substantial influence at the 36 km resolution, while its effect diminishes at the 3 km resolution. Conversely, influence of DDM_SNR is relatively low at the 36 km scale, but shows an increase at the 3 km resolution.

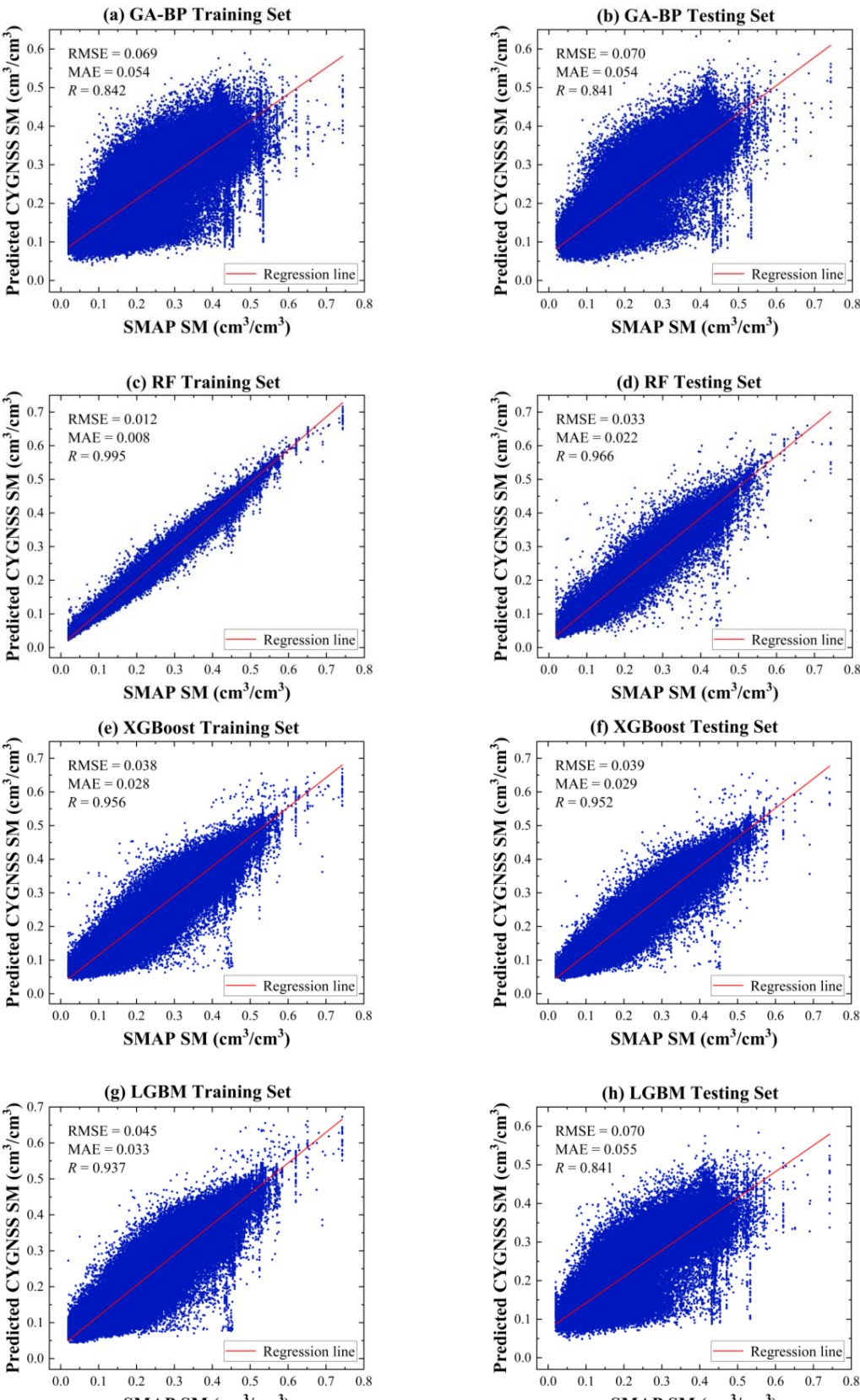

**Figure 6.** The retrieval accuracy of the four models in the training and testing datasets.

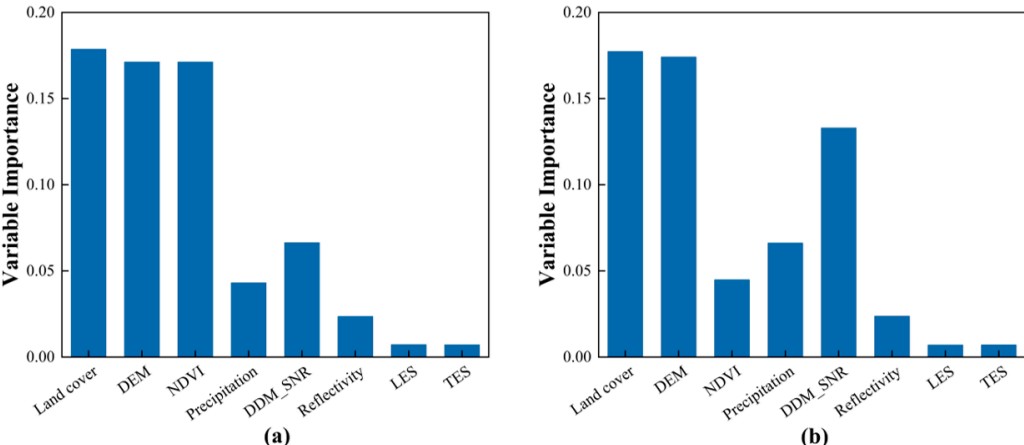

**Figure 7.** Variables' importance scores at (**a**) 36 km and (**b**) 3 km of XGBoost model.

*4.2. Assessing the Accuracy of Downscaled Soil Moisture Using In Situ Observations*

In the study area, we utilized the spatial coverage of CYGNSS and selected 78 sites from the ISMN with ground-based observation data as our research subjects from 1 September to 31 December 2019. These sites mainly come from SCAN, USCRN, and SNOTEL. Given the utilization of SMAP SM as a reference for the downscaling model in this study, it is essential to ensure the credibility of the assessment between downscaled SM and in situ SM observations. To achieve this, we initiated accuracy statistics for SMAP SM and in situ SM observations. Furthermore, to conduct a comprehensive time series analysis, we randomly selected four in situ sites for comparative evaluation with SMAP SM.

Table 4 delineates the comparison between in situ SM observations and corresponding SMAP SM. Analysis reveals that, out of all the in situ sites, 48 exhibit an MAE below 0.6, while 58 showcase an RMSE below 0.7. Additionally, 50 sites demonstrate an *R* exceeding 0.7. The respective average values for MAE, RMSE, and *R* stand at 0.051, 0.062, and 0.813. Overall, the majority of in situ sites exhibit commendable accuracy, thereby validating the reliability of the downscaling model constructed with SMAP SM as a reference. Figure 8 further illustrates the time series comparison of in situ SM observations and SMAP SM for randomly selected four in situ sites, with the time frame matching the dates of the downscaling model's prediction set. Of note is the 2–3-day revisit period of the SMAP satellite, which inhibits the guarantee of simultaneous coverage for each in situ site within the study area. Despite this limitation, the temporal variation of in situ SM observations (the blue line) closely aligns with that of SMAP SM (the red line). This alignment underscores SMAP SM's capability to capture the temporal dynamics of in situ SM, thereby validating the rationale for utilizing in situ SM observations in the downscaled SM assessment. To provide a quantitative assessment of the downscaled SM from the XGBoost model, Table 5 includes the accuracy statistical data for the downscaled SM and in situ SM observations.

According to the data analysis results in Table 5, for the total of 78 sites we studied, 62% of the sites, or about 49 sites, have an *R* greater than 0.600. This value is quite high, indicating that the downscaling model for these sites have good predictive performance. Similarly, we have 54% of sites, about 43, with an RMSE less than 0.070, which also indicates that these sites have a small retrieval error. Further, 53% of sites, or about 42 sites, have an MAE less than 0.060, indicating that our model has a high accuracy of retrieval for these sites. Overall, the average *R*, RMSE, and MAE of all sites are 0.712, 0.065, and 0.058, respectively, demonstrating the excellent performance of our model overall. However, we also found that the type of land cover may affect the accuracy of site validation. To gain a deeper understanding of this issue, we conducted further analysis. After analysis, we concluded that the downscaled SM for most sites using the XGBoost model is reliable compared to the in situ SM observations. However, the validation accuracy of a few sites is relatively poor. To better understand the accuracy of the downscaling model, we also considered the type of land cover at the site location, as shown in Figure 9. This means that

the type of land cover at the site location may affect the accuracy of the model. Through further research and model adjustment, we hope to better predict and understand this impact to optimize our model accuracy.

**Table 4.** Accuracy statistics for SMAP SM and in situ SM observations.

| Evaluation Index | Ranges | Number of In Situ Site | Average Value |
|---|---|---|---|
| MAE | <0.04<br>0.04–0.06<br>0.06–0.08<br>>0.08 | 38<br>10<br>20<br>10 | 0.051 |
| RMSE | <0.06<br>0.06–0.07<br>0.07–0.08<br>>0.08 | 48<br>10<br>11<br>9 | 0.062 |
| *R* | <0.60<br>0.60–0.70<br>0.70–0.80<br>>0.80 | 22<br>6<br>10<br>40 | 0.813 |

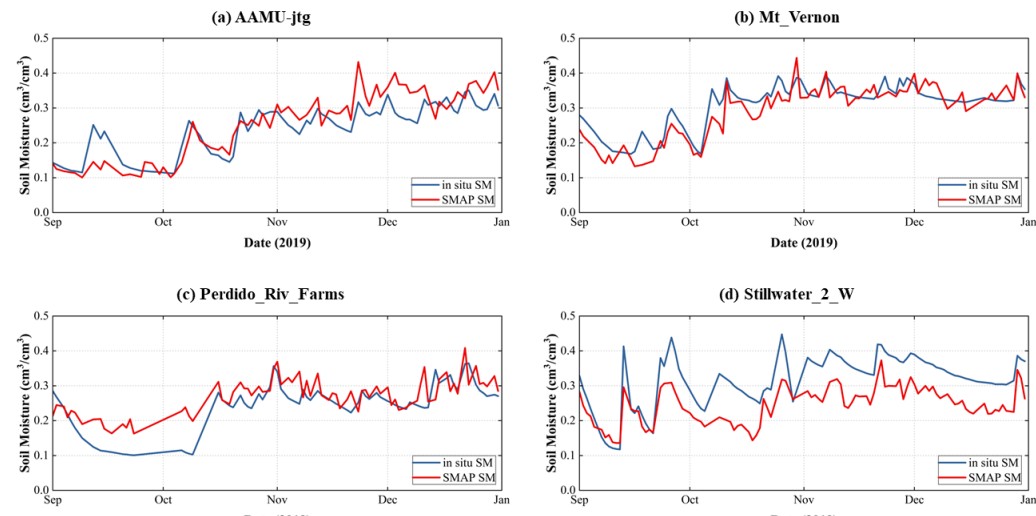

**Figure 8.** Time series of the SMAP SM and the in situ SM observations at the four sites.

**Table 5.** Accuracy statistics for downscaled SM and in situ SM observations.

| Evaluation Index | Ranges | Number of In Situ Site | Average Value |
|---|---|---|---|
| MAE | <0.04<br>0.04–0.06<br>0.06–0.08<br>>0.08 | 24<br>18<br>19<br>17 | 0.058 |
| RMSE | <0.06<br>0.06–0.07<br>0.07–0.08<br>>0.08 | 37<br>5<br>11<br>25 | 0.065 |
| *R* | <0.60<br>0.60–0.70<br>0.70–0.80<br>>0.80 | 29<br>9<br>7<br>33 | 0.712 |

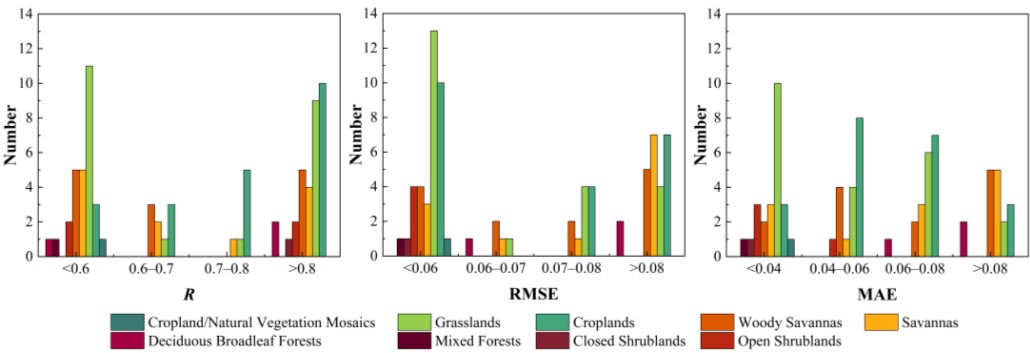

**Figure 9.** Precision statistics of in situ observations of different land cover types.

As seen in Figure 9, in the 78 in situ sites, 22 are located in grassland areas. Among these grassland sites, 14 have an *R* value less than 0.600, 13 have an RMSE less than 0.060, and 14 have an MAE less than 0.06. For the 21 sites situated in farmland areas, 18 have an *R* value less than 0.600, 10 have an RMSE less than 0.060, and 11 have an MAE less than 0.060. Of the 12 savanna sites, 7 have an *R* value less than 0.600, 3 have an RMSE less than 0.060, and 4 have an MAE less than 0.060. In the woody savannas, there are 13 sites, with 8 having an *R* value less than 0.600, 6 with an RMSE less than 0.060, and 4 with an MAE less than 0.060. Lastly, in the open shrublands, there are 4 sites. Two of these have an *R* value less than 0.600, all 4 have an RMSE less than 0.060, and 3 have an MAE less than 0.060. For the land cover types of deciduous broadleaf forests, mixed forests, closed shrublands, and cropland/natural vegetation mosaics, the number of sites are 3, 1, 1, and 1, respectively. Correspondingly, the sites with an *R* greater than 0.600 are 2, 0, 1, and 0. Sites with an RMSE less than 0.060 are found to be 0, 1, 1, and 1, while those with an MAE less than 0.060 are also 0, 1, 1, and 1.

This study primarily investigates the accuracy of downscaled SM models obtained through the application of the XGBoost model across nine different land cover types. The results indicate that sites located in grasslands and farmlands exhibit higher accuracy. This may be attributed to the fact that SM retrieval based on GNSS-R technology tends to be more accurate in flat areas than in areas with significant surface undulations or tree cover. Additionally, grasslands and farmlands are common land use types; hence, we have more sites for observation and validation. Conversely, the other seven land cover types have fewer sites, leading to a lack of sufficient validation data, which could be a significant factor affecting accuracy. Furthermore, we believe that other potential factors might influence the accuracy of SM retrieval. For instance, the varying soil properties and complexities across different regions could impact model performance. Highly heterogeneous soils or areas with significant rock content could lead to inaccurate predictions. Changes in precipitation and meteorological conditions might also affect the accuracy of the model. Prolonged droughts or consistent rainfall could potentially lead to decreased model performance during specific periods. The quality of GNSS-R technology data, the calibration process, and observational errors could impact model accuracy to some extent. Additionally, if there are changes in land use or land cover types in the study area during the observation period, this could affect the training and validation data, consequently influencing the model's accuracy. However, despite lower accuracy in some areas, it is evident from our results (Figure 9 and Table 5) that the downscaled SM model constructed in this study generally achieves satisfactory results. This suggests that our method exhibits adaptability and robustness, providing high accuracy in most scenarios.

### 4.3. Graphical Assessment of Spatial Distribution of Downscaled Soil Moisture

After the downscaled SM was validated using in situ sites, we proceeded to conduct a spatial analysis to evaluate the effectiveness of the downscaling approaches. The downscaled SM from the XGBoost model was visually compared with high-resolution MODIS EVI and MODIS ET products. The Enhanced Vegetation Index (EVI) is an indicator used

to assess and monitor the health and growth status of vegetation [55]. When SM is low, vegetation may be constrained by water availability, leading to slowed or stressed plant growth. This may manifest as lower EVI values. Conversely, when SM is high, plants may have ample water supply, promoting growth and resulting in higher greenness and elevated EVI. Evapotranspiration (ET) refers to the sum of evaporation from the land surface and transpiration from plants [56]. When SM is high, there is ample water supply in the soil, and plant roots can absorb sufficient water for transpiration, thereby promoting the ET process. Higher SM typically results in higher ET. Conversely, when SM is low, the water supply in the soil decreases and plants face water limitations, leading to reduced plant transpiration. Lower SM typically results in lower ET. Therefore, examining the variations in EVI and ET within the study area can indirectly reflect changes in SM.

The following sections compare the relationships among the downscaled SM, EVI, and ET for four periods: 9–14, 10–16, 11–17, and 12–19. We processed the downscaled SM from the XGBoost model using simple Kriging interpolation, and then conducted a spatial analysis with MODIS EVI and MODIS ET products.

As seen in Figure 10, the EVI values in the central and eastern of the study area are relatively high, while those in the northwest and southwest are lower. This is related to the vegetation cover in the study area and is consistent with the geographical characteristics of the study area described in Section 2.1. Compared with the downscaled SM and EVI at the same time, we can observe that areas in the study region with higher SM also have higher EVI values, such as the sides of the Central Valley in the middle and the Appalachian Mountains in the east. Conversely, areas with lower SM also have lower EVI values, such as in the western regions of Oklahoma and Salt Lake City. It can be proved that there is a correlation between SM and EVI. As seen in Figure 10(a-1), on September 14, the SM values in the Homochitto National Forest, the Sabine National Forest in the south-central study area, and the southeastern region are relatively high. Comparing this with Figure 10(a-3) at the same time, we can see that the ET values in these areas are also high. The same pattern can be observed when comparing Figure 10(b-1) with Figure 10(b-3), Figure 10(c-1) with Figure 10(c-3), and Figure 10(d-1) with Figure 10(d-3). SM, as one of the main sources of water for ET, may lead to higher ET in areas with higher SM. However, it is worth noting that there are some discrepancies in the spatial distribution of downscaled SM, EVI, and ET in some areas in the south-central study area (areas within the red box in Figure 10). Because the downscaling model is established at a 36 km grid scale, some extreme values are smoothed during the spatial aggregation process. As a result, the training samples selected in the model construction process are all smooth data, with fewer extreme values. This is not unique to our study, as all existing downscaling methods necessitate calibration with coarse-resolution data initially, making the aggregation of high-resolution predictors inevitable [26]. The result is that the downscaled SM has some mistakes.

Overall, the spatial distribution and temporal variation of the downscaled SM product generated in this paper are relatively consistent with EVI and ET, both of which have a certain correlation with SM. Therefore, this indirectly verifies the accuracy of the downscaled SM for the retrieval of SM in the study area.

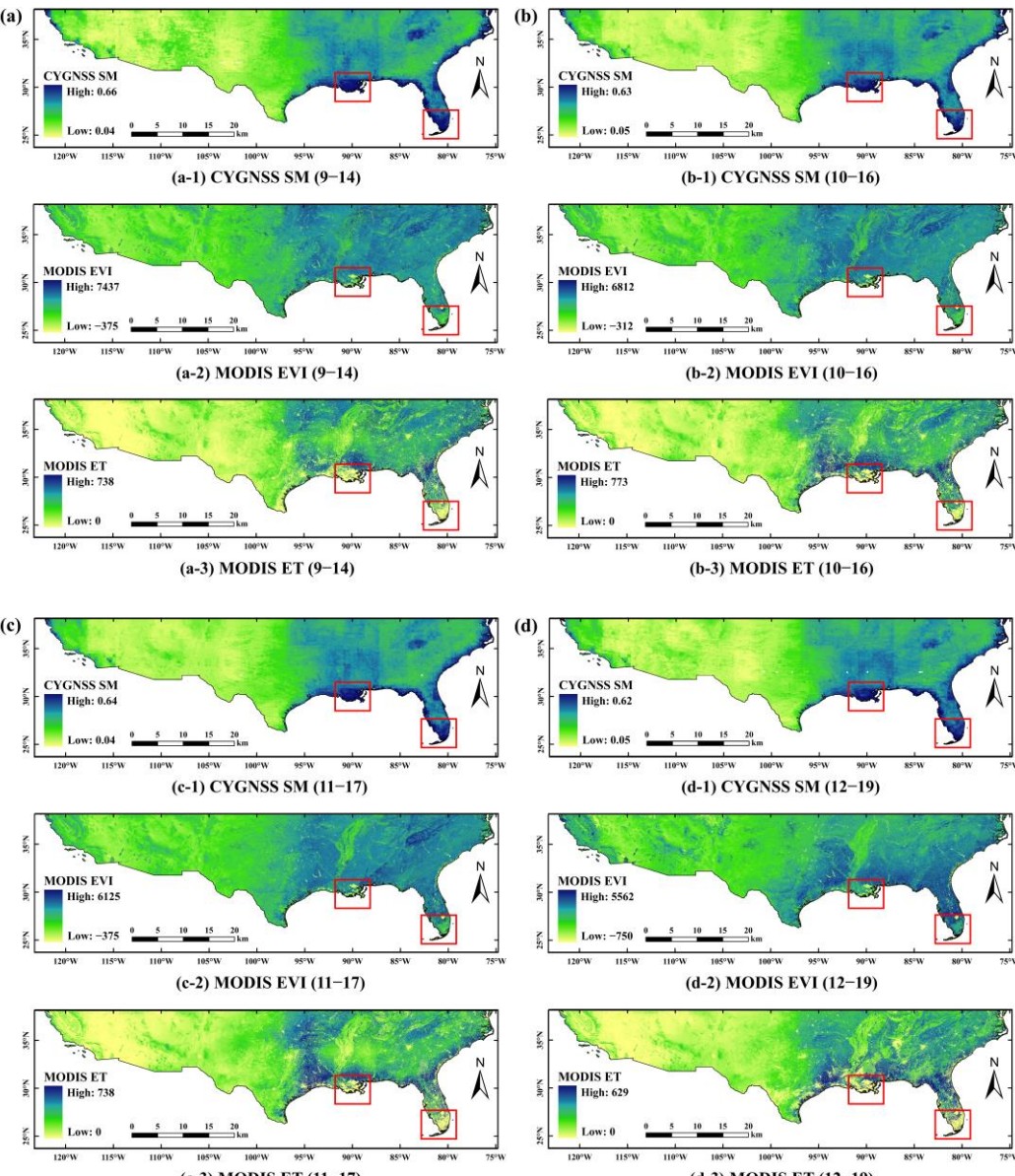

**Figure 10.** Distributions of the downscaled CYGNSS SM, MODIS EVI, and MODIS ET on 14 September 2019, 16 October 2019, 17 November 2019, and 19 December 2019.

## 5. Discussion

A key advantage of this study is the downscaling of CYGNSS based on the XGBoost model using L-band passive microwave SM (i.e., SMAP SM) and auxiliary variables. Instead, most previous studies downscale satellite SM products (AMSR-E, SMOS, and SMAP, etc.) based on optical data [39,53,54,57]. Another advantage is that it improves upon previous research that used CYGNSS to retrieve SM with a maximum spatial resolution of 9 km. Through the method of downscaling, this study has increased the spatial resolution of SM retrieval using CYNGSS to 3 km. Furthermore, the downscaled SM can more finely represent the spatial distribution changes in SM, offering substantial potential for applications such as irrigation planning in agriculture.

The noteworthy limitations of this study may present opportunities for improving the spatial downscaling of satellite SM outputs with coarse resolution. First, CYGNSS observables are collected at pseudo-random positions, with irregular spatial and temporal resolution. This is different from conventional remote sensing technologies, which have repeatable swaths and consistent local collection times. As a result, mapping CYGNSS

observables regularly in space presents a challenge in terms of assigning appropriate spatial grid sizes. The spatial resolution of CYGNSS observations can vary greatly, ranging from the first Fresnel zone with coherent reflections (0.5 km) to the scintillation zone with incoherent reflections (i.e., dozens of km). Traditional methods of mapping using regular spatial grids and integral time-step cannot fully account for this complexity in the spatiotemporal resolution of CYGNSS signals [58]. However, a transformation procedure is needed in order to match CYGNSS observables with other remote sensing and modeling data. This conversion process can introduce inaccuracies into the reflectivity, which could have implications, not just for this study, but for all similar research endeavors as well [12,17,59,60].

Second, during the model building process, the input CYGNSS observables and auxiliary variable are aggregated from high resolution to 36-km coarse resolution using a simple arithmetic averaging method. Furthermore, the SMAP SM encapsulates an average representation of SM, which is spread across a spatial resolution of 36 km. The average SM represents the SM of the whole region, and most of the information is ignored due to the coarse resolution. Hence, the training samples chosen during the model building process are smooth data with minimal extreme values. Models built from these samples invariably influence the downscaled SM. The scale discrepancy between the input data for model training and the SMAP products somewhat constrains the selection of suitable data during the regression model construction process. If a large amount of training data is necessary, choosing a research area that is large enough to assure the collection of enough training samples becomes crucial. During the application of the downscaling model, due to the increased heterogeneity and richer data representation at a 3 km resolution, there might be extreme values that were not encountered during model training. This corresponded with the results of Wakigari et al. [54]. Therefore, in practical applications, the downscaled SM has some inevitable errors. These errors are not randomly generated, but are closely related to the variance of SM in our training samples. In other words, the greater the degree of variation or dispersion of SM in the training samples, the greater the retrieval error may be. This is because a large variance means that the SM values in the dataset have greater changes, which may lead to more errors in the model's predictions. At the same time, the results of the downscaling process are significantly affected by the number and representativeness of the training samples. A sufficient number of training samples can provide more comprehensive information, helping the model to better learn and understand the characteristics of the data. The degree of representativeness of the samples directly affects the generalization ability of the model. If the samples can fully represent the characteristics of the entire data, then the model's retrieval results on unknown data will be more accurate.

Third, we used in situ SM observations, which are direct measurements from specific locations. However, these data may introduce some uncertainties when validating our downscaled SM model, mainly due to scale discrepancies. In our model, the downscaled SM represents an average SM value over a 3 km × 3 km area, which is a broader spatial average than in situ measurements. However, due to geographical conditions and human activities, there may be significant variations in SM within this area. For instance, if a location is under irrigation, it could lead to the recorded SM value at this point being much higher than the area's average. This scale discrepancy could pose some issues during the validation phase. For example, if a site is located in an irrigation area, its SM measurement might be significantly higher than the average SM of the area, leading to a large deviation between the SM measurement and the model retrieval at this site during model validation. This deviation is not a problem with the model, but is caused by spatial scale differences.

Fourth, the input optical data NDVI is inevitably affected by clouds during the model construction and model application. This also leads to the downscaled SM exhibiting optical properties. Factors such as cloud cover can impact the downscaled SM, leading to the occurrence of null values [39]. The presence of clouds may influence the availability of downscaled SM at the corresponding location. Furthermore, the 3 km resolution may

present challenges, potentially leading to the presence of missing values in the downscaled SM due to its inability to cover all processed pixels. To address this issue, we adopted an approach similar to that described by Wei Shangguan et al. [53] to fill these gaps. Specifically, we performed Kriging interpolation on the 3-km downscaled CYGNSS SM. However, it is important to acknowledge that the utilization of interpolation unavoidably introduces certain errors. Thus, some inconsistencies will present themselves during the validation stage.

Fifth, we utilized only four auxiliary variables, namely rainfall, land cover type, DEM, and NDVI. It is crucial to adequately consider the spatial scale of CYGNSS observation data, SM reference data, and auxiliary variables for the accuracy of SM retrieval. Factors such as soil type (sand, loam, clay, etc.) affect soil water absorption and the capacity to minimize water loss, as well as surface temperature variations and water evaporation caused by wind speed. As discussed by Volkan Senyurek [16], soil texture features are considered to have the greatest impact on retrieval SM among auxiliary inputs. In summary, while the method proposed in this paper has achieved a commendable accuracy in SM retrieval, there is room for improvement by considering a wider range of auxiliary factors. This has the potential to further enhance the accuracy of SM retrieval using GNSS-R.

Sixth, in assessing the spatial distribution of downscaled SM, this paper has not yet considered the influence of a variety of factors on plant growth and ET. These factors include light conditions, temperature, soil texture, and carbon dioxide concentration, all of which may have an impact on the accuracy of MODIS EVI and MODIS ET products, as changes in these factors may result in changes in vegetation activity and ET. There are limitations in using these products to assess the spatial distribution of downscaled SM. Considering that these factors may add to the complexity of the assessment, future research could attempt to integrate these factors to obtain more accurate downscaled SM estimates.

Seventh, in the process of evaluating the spatial distribution of downscaled SM, we employed the Kriging interpolation method. However, the Kriging interpolation method might not be optimal, as it measures SM with limited physical significance and could result in spatial heterogeneity. Therefore, in future research, it is essential to compare different interpolation methods and investigate their impact on downscaled SM. Selecting the most suitable interpolation method will facilitate the assessment of the spatial distribution of downscaled SM.

Finally, there are some limitations concerning the geographical scope of our study area and the duration of the data utilized. When validating the downscaled SM using in situ sites, we observed that, aside from grasslands and farmlands, the availability of in situ sites for other land cover types was limited. This paucity of data can hinder a comprehensive validation. By expanding the study area, the number of in situ sites for other land cover types would increase, thereby augmenting the validation dataset and enhancing the accuracy and reliability of our model performance assessment. In this study, the SM downscaling model was constructed using data from January to August 2019, while data from September to December was used for SM retrieval. This may introduce seasonal biases into the constructed downscaling model, leading to certain inaccuracies. Extending the data period for a year or even longer could mitigate such seasonal effects, thus boosting the reliability of the downscaling approach.

## 6. Conclusions

In this paper, we propose a downscaling method for CYGNSS SM based on the XGBoost algorithm, using high-resolution CYGNSS observables and auxiliary variables as input data to improve the spatial resolution of GNSS-R technique retrieval of SM to 3 km. The method selects common downscaled variables such as DEM, land cover, NDVI, and rainfall. We enhance and improve the polynomial-based downscaling regression model by incorporating parameters of SR, SNR, LES, and TES from CYGNSS. Experiments were conducted using data covering the southern United States, and the results were validated by 78 in situ sites. The results show that the downscaled SM achieves *R*, RMSE, and MAE

of 0.712, 0.068, and 0.058, respectively, compared with the in situ SM observations. Spatial analysis using MODIS EVI and MODIS ET products shows that the spatial distribution and temporal variation of the downscaled CYGNSS SM products are more consistent with the EVI and ET products. The feasibility of the method is proved. Additionally, we discuss a number of problems that came up throughout the downscaling and validation process.

Overall, the findings of this study offer valuable insights for enhancing SM downscaling methods. This is crucial for advancing high-resolution SM retrieval. In future research, it will be key to develop gap-filling methods to address missing remote sensing data and refine the downscaling model. Additionally, researchers could consider using satellite SM products from various sources (e.g., SMAP, SMOS, AMSR-E, NASA-USDA, etc.) as reference values for downscaling models. This could aid in determining the most efficient downscaled SM products that are best suited to the particular conditions of the selected study area. Additionally, future research could consider a gradual downscaling approach (for instance, downsizing from 36 km to 9 km, followed by a reduction from 9 km to 3 km), as opposed to an immediate downscaling from 36 km to 3 km.

**Author Contributions:** Q.L.: Methodology, Writing—original draft, Writing—review and editing. Y.L.: Methodology, Writing—original draft, Writing—review and editing. Y.G.: Visualization, Validation. X.L.: Supervision. C.R.: Project administration. W.Y.: Software. B.Z.: Data curation. X.J.: Investigation. All authors have read and agreed to the published version of the manuscript.

**Funding:** This work was supported by the National Natural Science Foundation of China (No. 41901409); the Natural Science Foundation of Guangxi (No. 2021GXNSFBA220046); the National Natural Science Foundation of China (No. 42064003).

**Data Availability Statement:** CYGNSS data can be downloaded from the Physical Oceanography Distributed Active Archive Center (PO.DAAC, https://podaac.jpl.nasa.gov, accessed on 1 April 2023). SMAP data can be obtained from the National Snow and Ice Data Center (NSIDC, https://nsidc.org, accessed on 15 April 2023). The in situ soil moisture data can be accessed publicly (http://ismn.geo.tuwien.ac.at, accessed on 20 April 2023).

**Acknowledgments:** The authors are grateful to NASA EOSDIS Physical Oceanography Distributed Active Archive Center (DAAC), Jet Propulsion Laboratory, Pasadena, CA, USA, for making the CYGNSS data available, at https://www.esrl.noaa.gov/psd/ (accessed on 1 April 2023).

**Conflicts of Interest:** The authors declare no conflict of interest.

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
