# Peer review of "Enhancing Spatial Resolution of GNSS-R Soil Moisture Retrieval through XGBoost Algorithm-Based Downscaling Approach: A Case Study in the Southern United States"

_remotesensing, doi:10.3390/rs15184576_

Round 1
Reviewer 1 Report
The article addresses the issue of estimating a high spatial resolution soil moisture product with CYGNSS data in synergy with other auxiliary data. This is an interesting study with the aim of improving the spatial resolution of GNSS-R products. Despite the quality of the proposed work, there are still essential comments to be clarified.
1) The processed period seems relatively short for a solid training of algorithms. I propose extending the treatment for at least one more year?
2) There are few details on information related to the number of measurements considered per day for each pixel?
3) The 3km resolution can pose a problem because with the CYGNSS data we do not necessarily cover all the processed pixels, there are surely holes for certain pixels?
4) Can there be a discussion regarding the effect of land cover on soil moisture estimation?
Author Response
Dear Reviewer,
I am submitting the revised version of my manuscript titled "Enhancing Spatial Resolution of GNSS-R Soil Moisture Retrieval through XGBoost algorithm - Based Downscaling Approach: A Case Study in the Southern United States" along with this cover letter, in response to the invaluable feedback provided by the reviewers. I am genuinely grateful for the thoughtful evaluation and constructive suggestions, which have significantly contributed to the enhancement of the quality of my paper.
In the submitted document, I provide a detailed response to each reviewer's comment and outline the specific revisions made in the manuscript to address these points. I am confident that the modifications and clarifications I have implemented will improve the accuracy, clarity, and overall compliance of the paper with the journal's standards and guidelines.
A few key highlights from the revisions include:
- Addressing each question and suggestion raised by the reviewers and incorporating corresponding changes throughout the manuscript.
- Ensuring that aspects such as data, methodology, and results are thoroughly elucidated to enhance reader understanding.
- In response to the query posed by the reviewer regarding, I conducted an in-depth analysis and provided relevant empirical evidence to support my conclusions.
It is an honor to have the opportunity to contribute to Remote Sensing, and I am committed to collaborating with you and the editor to ensure that the paper reaches the highest standard. If you have any inquiries about the modifications or responses I have provided, or if further information is needed, please do not hesitate to contact me. I eagerly anticipate your evaluation of the revised manuscript.
Once again, I appreciate your attention and support for my work.
Sincerely,
Qidi Luo

Reviewer 2 Report
The manuscript presents a valid approach to perform pixel downscaling using combined optical and gnss-r, trained using GNSS-R data at different resolutions.
The application of the model is clear, and sufficient figures are provided to understand the process followed by the authors.
However, a few considerations shall be taken into account before publication:
1. SMAP data over areas with VWC > 5 km/m2 are not valid/have large errors, did the authors filter for those data points?
2. For the sake of comparability, the R, MAE, and RMSE of SMAP SM @ 36 km over the in situ SM points should be provided.
3. It would be great to explore the "weight" provided to the model from each variable. I'd propose some strategy similar to what is proposed in https://doi.org/10.1016/j.rse.2021.112801 would be great. Comparing the effect of adding TES/LES, or adding the whole GNSS-R dataset, or the ET, NDVI, etc. would be great for assessment for future SM products.
Author Response

(The authors gave the same response as above.)

Reviewer 3 Report
This is a well-written English article. The study introduces a downscaling method for GNSS-R and SMAP soil moisture data fusion, which establishes a nonlinear relationship among CYGNSS observables, auxiliary variables and 36km resolution SMAP soil moisture products, and employs the XGBoost algorithm to develop a downscaling retrieval model for soil moisture. The study fully leverages the high spatiotemporal resolution benefit of GNSS-R, and makes a novel contribution to enhancing the spatial resolution of soil moisture retrieval using GNSS-R technology. Please read carefully the following suggestions to improve your manuscript.
2. Materials and Methods
Line 171 presents various methods to estimate the surface reflectance, line 172 indicates that this study adopts the method proposed by Rodriguez-Alvarez et al. to compute the reflectance. I suggest that these two sentences should be supplemented with the merits of the method in Rodriguez-Alvarez et al. and the rationale for selecting this method, to bridge these two sentences.
Line 199, where the research data period is specified, is prone to be ignored by the reader. I suggest incorporating the research period in the second paragraph of section 2.2, and clarifying the period together with the data products employed will be more fluent.
Line 244, I have a query, besides the influencing factors discussed in the text, I also considered other influencing factors, such as: soil type (sand, loam, clay, etc.) whether it will affect its capacity to absorb water and minimize water loss; surface temperature variations, water evaporation induced by wind speed and so forth. If you find these aspects relevant, please include them in the discussion section.
3. Soil Moisture Downscaling Framework
Line 377 should not be indented.
4. Results
In section 4.2, for some sites with inferior evaluation metrics, the potential causes can be examined in detail, and these causes may suggest further improvements for the model.
Some of the images in the text have low resolution and appear very fuzzy, such as Figure 4, Figure 6.
The fonts of the images in the text are not uniform, and some fonts are even illegible due to their small size, please revise.
Line 560 states that there are fewer sites of other seven land cover types, leading to inadequate validation data. This influencing factor can be addressed by enlarging the research area. With the expansion of the research area, the corresponding sites of other seven land cover types will also grow, enhancing the validation data and increasing the accuracy.
Employing EVI and ET to assess the efficacy of the downscaling method is a standard method, but the growth of plants is influenced by many factors, such as light conditions, temperature conditions, soil conditions, carbon dioxide concentration and so forth. Please clarify how to control for these influencing factors, if they are limitations, they can be discussed in the discussion section to enhance the persuasiveness of the article.
Minor editing of English language required
Author Response

(The authors gave the same response as above.)

Round 2
Reviewer 1 Report
Despite the improvement of the article, authors don't answer all comments.
1) The number of daily CYGNSS samples is not clearly precised.
2) The use of a kriging approach for soil moisture is not physically evident. The spatial hetoregeneity could be very important . This needs to be clearly identified in discussion section.
Author Response

(The authors gave the same response as above.)

Reviewer 3 Report
It is good for this manuscript.
Author Response

(The authors gave the same response as above.)
